# Aligned Electrospun PCL/PLA Nanofibers Containing Green-Synthesized CeO_2_ Nanoparticles for Enhanced Wound Healing

**DOI:** 10.3390/ijms26136087

**Published:** 2025-06-25

**Authors:** Yen-Chen Le, Wen-Ta Su

**Affiliations:** Department of Chemical Engineering and Biotechnology, National Taipei University of Technology, 1, Sec. 3, Chung-Hsiao E. Rd, Taipei 106344, Taiwan

**Keywords:** *Hemerocallis citrina* extract, green synthesis, cerium oxide nanoparticle, aligned electrospinning, wound healing

## Abstract

Wound healing is a complex biological process that benefits from advanced biomaterials capable of modulating inflammation and promoting tissue regeneration. In this study, cerium oxide nanoparticles (CeO_2_NPs) were green-synthesized using *Hemerocallis citrina* extract, which served as both a reducing and stabilizing agent. The CeO_2_NPs exhibited a spherical morphology, a face-centered cubic crystalline structure, and an average size of 9.39 nm, as confirmed by UV-Vis spectroscopy, FTIR, XRD, and TEM analyses. These nanoparticles demonstrated no cytotoxicity and promoted fibroblast migration, while significantly suppressing the production of inflammatory mediators (TNF-α, IL-6, iNOS, NO, and ROS) in LPS-stimulated RAW264.7 macrophages. Gene expression analysis indicated M2 macrophage polarization, with upregulation of Arg-1, IL-10, IL-4, and TGF-β. Aligned polycaprolactone/polylactic acid (PCL/PLA) nanofibers embedded with CeO_2_NPs were fabricated using electrospinning. The composite nanofibers exhibited desirable physicochemical properties, including porosity, mechanical strength, swelling behavior, and sustained cerium ions release. In a rat full-thickness wound model, the CeO_2_ nanofiber-treated group showed a 22% enhancement in wound closure compared to the control on day 11. Histological evaluation revealed reduced inflammation, enhanced granulation tissue, neovascularization, and increased collagen deposition. These results highlight the therapeutic potential of CeO_2_-incorporated nanofiber scaffolds for accelerated wound repair and inflammation modulation.

## 1. Introduction

Skin wounds result from lacerations, fractures, burns, or underlying pathological conditions. The wound healing process is a complex, dynamic, and multi-phase biological cascade that includes hemostasis, coagulation, inflammation, cellular migration and proliferation, angiogenesis, maturation, collagen deposition, granulation tissue formation, and eventual tissue remodeling [1,2]. Impaired or delayed wound healing can lead to severe complications such as infection, tissue necrosis, limb loss, or even death. Therefore, timely and effective wound management is essential to promote healing, prevent infection, and reduce the risk of complications.

An ideal wound dressing should exhibit biocompatibility, non-toxicity, appropriate breathability, moisture retention, exudate absorption, and non-adhesiveness to prevent pain and secondary injury during dressing changes [3]. Recent advances in wound dressing technologies have led to the development of various biomaterials, including films, hydrogels, electrospun nanofibers, and three-dimensional (3D)-printed scaffolds [4]. These materials are typically biodegradable, biocompatible, and porous, providing a favorable microenvironment for cell attachment, proliferation, migration, and re-epithelialization. The incorporation of bioactive compounds can further enhance wound healing by reducing inflammation, promoting tissue regeneration, and minimizing scab formation [2].

Electrospun nanofiber scaffolds, with their 3D porous architecture, have demonstrated the ability to accelerate wound healing [5]. Electrospinning enables the fabrication of nanofibers in various morphologies, including random, aligned, and coaxial configurations, using biodegradable polymers [6]. These nanofibers closely mimic the extracellular matrix (ECM) in both physical and mechanical properties, supporting cell proliferation, migration, angiogenesis, and tissue regeneration [5]. Aligned nanofibers, in particular, have been shown to enhance cell attachment and directional migration, making them more suitable for wound dressing applications than randomly oriented fibers [7,8,9].

Poly(ε-caprolactone) (PCL) and poly(lactic acid) (PLA) are synthetic, biodegradable, biocompatible, and bioresorbable polyesters approved by the U.S. Food and Drug Administration (FDA) for biomedical applications. When electrospun into nanofibers, these polymers form a structure analogous to the ECM, with high surface area, porosity, small pore size, and suitable mechanical strength, rendering them effective platforms for tissue engineering and regenerative medicine [10,11]. Despite both being linear aliphatic polyesters, PCL degrades more slowly than PLA due to differences in molecular architecture, while PLA exhibits a higher Young’s modulus and tensile strength. As such, electrospun PCL/PLA-blended nanofibers offer a balanced combination of mechanical performance and controlled degradation, making them ideal for wound dressing applications [12].

Nanomaterials possess unique physicochemical properties, such as small size, a high surface area-to-volume ratio, and tunable surface functionality, which make them highly attractive for biomedical applications, including wound healing. Cerium (Ce), the most abundant rare earth element in the lanthanide series, has been shown to modulate inflammation, promote cellular proliferation and migration, and enhance differentiation processes [13]. Cerium oxide nanoparticles (CeO_2_NPs) exhibit potent antioxidant and anti-inflammatory activities by scavenging reactive oxygen species and regulating redox-sensitive signaling pathways. These properties contribute to improved cell proliferation, tissue regeneration, and repair, thereby supporting their application in wound healing therapies [14,15,16].

*Hemerocallis citrina* (*H. citrina*), a perennial herbaceous plant, is rich in bioactive compounds such as polyphenols, flavonoids, alkaloids, and anthraquinones. It is traditionally used for its antioxidant, antibacterial, and anti-inflammatory properties in the treatment of skin injuries and burns [17,18]. Compared to conventional physical and chemical synthesis methods, green synthesis of metal oxide nanoparticles using plant-derived secondary metabolites offers a sustainable alternative. This approach utilizes natural reducing and stabilizing agents, is energy-efficient, cost-effective, environmentally friendly, and avoids the use of toxic organic solvents [19,20].

In this study, we aim to synthesize CeO_2_NPs using *H. citrina* extract via a green synthesis approach. The synthesized nanoparticles were characterized by UV-Vis spectroscopy, scanning electron microscopy (SEM), transmission electron microscopy (TEM), energy-dispersive X-ray spectroscopy (EDS), selected area electron diffraction (SAED), X-ray diffraction (XRD), and Fourier transform infrared spectroscopy (FTIR). These nanoparticles were incorporated into a PCL/PLA solution and electrospun into aligned nanofibers. We evaluated their anti-inflammatory effects and ability to promote cell migration in vitro, followed by an in vivo assessment of their wound healing efficacy using a rat excisional wound model. The study aims to demonstrate the biomedical potential of aligned electrospun nanofibers embedded with green-synthesized CeO_2_NPs as a promising wound dressing material.

## 2. Results

### 2.1. Optimization of H. citrina Extract Yield

Microwave-assisted extraction was performed at power levels ranging from 250 W to 400 W, while maintaining a fixed extraction time of 25 min and a solid-to-liquid ratio of 1:4 (*w*/*v*). As shown in Figure 1A, the highest total polyphenol content (75.31 mg GAE/g) was obtained at 350 W. Subsequently, various extraction times were evaluated under fixed conditions of 350 W power and the same solid-to-liquid ratio. The total polyphenol yield reached its maximum (79.27 mg GAE/g) at 30 min (Figure 1B). Therefore, the optimal extraction conditions were determined to be 350 W and 30 min. Under these optimized parameters, the total flavonoid content of the extract was measured at 28.97 mg QE/g.

### 2.2. Physicochemical Characterization of CeO_2_NPs

CeO_2_NPs exhibit a characteristic absorption band due to surface plasmon resonance, typically observed around 350 nm [21]. The UV-Vis absorption spectrum of CeO_2_NPs synthesized using *H. citrina* extract displayed a prominent peak at approximately 342 nm (Figure 2A), confirming the successful formation of CeO_2_NPs. The FTIR spectra (Figure 2B) revealed that the *H. citrina* extract exhibited absorption peaks at 3317 cm^−1^ (O–H), 2923 cm^−1^, and 2853 cm^−1^ (C–H stretching of methyl and methylene groups), 1611 cm^−1^ (C=O), and 1017 cm^−1^ (C–O stretching). In contrast, CeO_2_NPs showed characteristic peaks at 538 cm^−1^ and 622 cm^−1^, corresponding to Ce–O vibrations [22], and a minor peak at 1010 cm^−1^ possibly attributable to residual organic moieties [23].

SEM revealed the sheet-like surface morphology of the CeO_2_NPs (Figure 2C), indicative of aggregation due to high surface energy. EDS confirmed the elemental composition, showing cerium (34.44%) and oxygen (42.63%) (Figure 2D). TEM images (Figure 2E) showed spherical to sheet-like particles with an average diameter of 9.388 nm (Figure 2F), as measured using ImageJ (Version 1.5.3, National Institutes of Health, Bethesda, MD, USA). XRD analysis (Figure 2G) demonstrated reflection peaks at 2θ values corresponding to the (111), (200), (220), (311), (222), (400), (331), and (420) planes of face-centered cubic CeO_2_, consistent with JCPDS Card No. 81-0792 [24]. SAED patterns (Figure 2H) showed distinct rings and spots, affirming the crystalline nature of the nanoparticles.

The antioxidant activity of CeO_2_NPs was evaluated using the DPPH assay (Figure 2I). The IC_50_ values were 65.69 μg/mL for ascorbic acid, 503.47 μg/mL for *H. citrina* extract, and 319.12 μg/mL for CeO_2_NPs. At 1000 μg/mL, CeO_2_NPs exhibited 88.17% radical scavenging activity, indicating significant antioxidant potential relevant to wound healing applications.

### 2.3. Characterization of PCL/PLA/CeO_2_NP Nanofibers

Figure 3A represents the morphology and diameter of all the different prepared nanofibers. SEM images of PCL/PLA, PCL/PLA/0.5% CeO_2_NPs, and PCL/PLA/1% CeO_2_NPs nanofibers demonstrated smooth, bead-free, and uniformly aligned morphologies, confirming successful fabrication via electrospinning. From the nanofiber diameter distribution curve, the average fiber diameters were 832.21 ± 193.71 nm, 713.89 ± 208.50 nm, and 643.42 ± 160.67 nm, respectively. The observed reduction in fiber diameter with increasing nanoparticle content is attributed to enhanced solution conductivity [25]. Mechanical testing revealed that pure PCL/PLA nanofibers exhibited superior tensile strength and Young’s modulus. The incorporation of CeO_2_NPs reduced mechanical integrity, likely due to disruption in polymer chain continuity (Figure 3B).

Water vapor transmission rates (WVTRs) for PCL/PLA, 0.5% CeO_2_NPs, and 1% CeO_2_NPs nanofibers were 2074.91, 2122.07, and 2216.38 g/m^2^·d, respectively (Figure 3C), all within the ideal range (2000–2500 g/m^2^·d) for wound dressings [26]. The porosity of pure PCL/PLA nanofibers and those containing 0.5% or 1% CeO_2_NPs were 58.74 ± 1.91%, 62.89 ± 0.62%, and 70.67 ± 0.97%, respectively. After the incorporation of 0.5% and 1% CeO_2_NPs, the degradation rates of the nanofibers increased by 19.98% and 23.06%, respectively (Figure 3D), and the swelling ratio increased from 222% to 245% and 286% (Figure 3E). This enhancement may be attributed to the disruption of the structural integrity of the PCL/PLA matrix by CeO_2_NPs, which increases the specific surface area of the nanofibrous membrane and reduces the crystallinity of PCL, thereby accelerating the degradation rate [27]. Consequently, the hydrophobicity of the PCL/PLA nanofibers was reduced, as evidenced by a decrease in water contact angle from 116.6 ± 3.2° to 103.6 ± 2.1° and 105.9 ± 1.6°, respectively. The release profiles of cerium ions (Figure 3F) demonstrated an initial burst followed by sustained release. By day 10, Ce ion concentrations released from 0.5% and 1% CeO_2_NPs nanofibers reached 4.15 ppm and 8.77 ppm, respectively, supporting their suitability for prolonged therapeutic application.

### 2.4. Biocompatibility, Anti-Inflammatory, and Antioxidant Properties of CeO_2_NPs

Cell viability assays showed that both 0.5% and 1% CeO_2_NPs maintained >82% viability in L929 and RAW264.7 cells after 3 days of culture (Figure 4A,B), confirming low cytotoxicity and excellent biocompatibility. In scratch assays (Figure 4C,D), 1% CeO_2_NPs significantly enhanced L929 fibroblast migration between the scratch of 200 μL pipette tip, increasing the migration rate 2.46-fold compared to control group.

LPS-induced inflammatory responses in RAW264.7 cells resulted in elevated TNF-α and nitric oxide (NO) levels. Pretreatment with 1% CeO_2_NPs reduced TNF-α by 49.0% (*p* < 0.01) and NO by 49.4%, while 0.5% CeO_2_NPs led to a 17.6% reduction in NO (Figure 5A,B). CeO_2_NPs also mitigated intracellular ROS production by 19% and 35%, respectively (Figure 5C). Moreover, LPS induction increased M1 macrophage markers (TNF-α, IL-6, IL-1β, iNOS) 3.1-, 3.9-, 14.0-, and 2.4-fold, respectively. Pretreatment with 1% CeO_2_NPs reduced these markers to 64.0%, 74.6%, 79.2%, and 81.5% of LPS-only levels (Figure 6A). In contrast, M2 markers (IL-10, IL-4, Arg-1, TGF-β) were significantly upregulated, 1.4-, 1.2-, 3.2-, and 1.5-fold, respectively, in CeO_2_NPs-treated groups (Figure 6B), demonstrating CeO_2_NPs’ ability to shift macrophage polarization from a pro-inflammatory (M1) to a reparative (M2) phenotype.

### 2.5. In Vivo Wound Healing Assessment

In vivo studies (Figure 7A,B) revealed that the PCL/PLA/1%CeO_2_NPs nanofiber dressing significantly enhanced wound closure, achieving a 94% closure rate by day 11—22% higher than the control group—with complete healing by day 14. H&E staining showed extensive granulation tissue formation (black lines), minimal inflammatory infiltration, and full re-epithelialization in the CeO_2_-treated group (Figure 7C). Quantitative analysis confirmed increased granulation tissue area and early neovascularization, with reduced vessel density by day 14 indicative of vascular maturation (Figure 7D).

Masson’s trichrome staining (Figure 7E) revealed abundant collagen deposition and well-organized epithelium, with greater epithelial thickness than control groups. The quantification of collagen deposition is shown in Figure 7F. Although the PCL/PLA group exhibited lower healing rates compared to the PCL/PLA/1%CeO_2_ group, it still performed significantly better than the sterile gauze control (*p* < 0.01, 9th to 14th day after surgery). This result collectively demonstrates that aligned nanofiber dressings incorporating CeO_2_NPs significantly accelerate wound healing and hold promise for clinical wound care applications.

## 3. Discussion

Nanoparticles exhibit unique physicochemical properties, including a large specific surface area, high biocompatibility, bioabsorbability, and the capacity to promote tissue regeneration, making them highly promising candidates for applications in regenerative medicine. For instance, gold nanoparticles have been utilized in photothermal therapy for prostate cancer [28], while silver nanoparticles are commonly employed as antimicrobial coatings on medical devices to prevent infection [29]. Zinc oxide nanoparticles serve as ultraviolet filters in cosmetic sunscreen formulations [30] and an inducer of osteogenic differentiation in periodontal ligament stem cells [31], and palladium nanoparticles have demonstrated cytotoxic effects by inducing apoptosis in HeLa cells [32]. These advancements underscore the transformative role of nanotechnology in developing novel therapeutic strategies within regenerative and biomedical sciences.

Among the various synthesis techniques, green synthesis has emerged as one of the most effective and environmentally sustainable approaches for the fabrication of nanoparticles. Compared to conventional physical and chemical methods, green synthesis offers numerous advantages, such as the use of natural reducing, capping, and stabilizing agents, lower production costs, reduced toxicity, improved biocompatibility, and enhanced nanoparticle stability [33,34]. Typically, this method employs plant extracts or microorganisms as bioreducing agents for the synthesis of metal-based nanoparticles. For example, *Crocus sativus* extract and *Streptomyces* sp. MSK03 fermentation broth have been successfully used to synthesize gold nanoparticles [35,36], while *Lallemantia royleana* leaf extract has been employed in the synthesis of silver nanoparticles [37].

In the present study, cerium ions were reduced by polyphenols and flavonoids present in *H. citrina* extract, followed by high-temperature calcination to produce face-centered cubic CeO_2_NPs with an average particle size of approximately 9 nm, as confirmed by UV-Vis spectroscopy and FTIR. Furthermore, CeO_2_NPs exhibited significantly greater DPPH free radical scavenging activity compared to *H. citrina* extract and demonstrated antioxidant activity comparable to that of ascorbic acid. These findings suggest that CeO_2_NPs possess strong anti-inflammatory and wound healing potential [38]. Their redox-active nature, particularly the presence of mixed valence states and oxygen vacancies, enables CeO_2_NPs to act as self-regenerating antioxidants, mimicking the activity of enzymes such as superoxide dismutase [16]. As natural antioxidants, they reduce the infiltration of macrophages and neutrophils in inflamed tissues [39]. Given the critical role of oxidative stress and inflammation in delayed wound healing, their modulation is essential for achieving efficient tissue repair.

In 2017, Serebrovska et al. reported that treatment with CeO_2_NPs in a rat model of pneumonia significantly reduced lung tissue damage, ROS generation in both lung and blood, and expression of pro-inflammatory cytokines such as TNF-α, IL-6, and CxCL2 [40]. Similarly, Yi et al. demonstrated that CeO_2_NPs mimic endogenous antioxidant enzyme activity, reduce oxidative stress, and suppress inflammatory mediator production, further highlighting their therapeutic potential in managing inflammatory disorders [41]. Consistent with these findings, the present study confirmed that pretreatment with CeO_2_NPs effectively inhibited LPS-induced production of TNF-α, NO, and ROS in RAW264.7 macrophages, demonstrating their potent anti-inflammatory and antioxidant properties. Moreover, CeO_2_NPs downregulated M1 macrophage-associated gene expression while upregulating M2-associated genes, supporting their ability to promote anti-inflammatory macrophage polarization. These results align with those reported by Ribera et al. [42]. CeO_2_NPs can induce angiogenesis by modulating the intracellular oxygen environment through the activation of HIF-1α and the Ref-1/APE1 signaling pathway [43]. Augustine et al. also demonstrated that CeO_2_NPs promote angiogenesis and facilitate wound healing in diabetic models [44]. Increasing evidence has shown that CeO_2_NPs enhance cell proliferation and migration, regulate multiple cellular signaling pathways, alleviate inflammatory responses and oxidative stress, and improve various endothelial cell functions, thereby accelerating the wound healing process [15,45]. These findings highlight the significant potential of CeO_2_NPs in the field of biomedicine [39,46].

Wound dressings play a vital role in protecting injured tissue by serving as a physical barrier against microbial invasion and reducing pain, thereby attenuating neuroinflammatory responses. An ideal wound dressing should be breathable and porous to prevent maceration, promote gas exchange, maintain optimal moisture levels, and support tissue regeneration. In recent years, advanced wound dressings have been designed not only to facilitate natural healing but also to promote angiogenesis and enhance the migration of fibroblasts and keratinocytes [47]. Electrospun nanofiber dressings offer a porous architecture that permits oxygen diffusion, retains moisture, and absorbs wound exudate [48]. Their structural and physicochemical resemblance to the natural ECM provides a three-dimensional environment conducive to cell adhesion, proliferation, differentiation, and migration [49]. Additionally, these nanofibers can modulate apoptosis, regulate the release of growth factors, and activate intracellular signaling pathways [50]. PCL and PLA or a mixture of them are widely utilized drug-releasing platforms in tissue engineering and biomedical applications due to their excellent biocompatibility and tunable degradation profiles. PCL/PLA nanofiber has high biocompatibility and low toxicity to cells or tissues, which is very suitable for drug delivery and clinical biomedical materials in regenerative medicine [51]. Hashem et al. utilized PCL/PLA nanofiber delivery of VEN to avoid hepatic metabolism and enzymatic degradation in the GIT and develop an effective control of drug release [52]; Stojanov et al. delivered vaginal lactobacilli for preventing and treating vaginal infections [53]; Castro et al. blended ZnO in PCL/PLA nanofiber for biomedical application [54]; and Hashem et al. incorporated HA with PCL/PLA nanofiber for bone healing applications [52]. In this study, Figure 3F shows that PCL/PLA nanofibers stably degrade and slowly release CeO_2_NPs, continuously providing wound healing agents, proving that PCL/PLA nanofibers are potential drug delivery media in regenerative medicine. The current development of wound dressings is progressively shifting towards hydrophilic dressings, multilayer foam dressings, autologous tissue-engineered skin, and artificial skin substitutes [55,56]. These innovations significantly reduce dressing change frequency and related costs while enhancing healing rates and patient comfort, particularly in chronic wounds. Recent advances not only reinforce passive protection but also introduce smart dressings that incorporate electrical stimulation, wireless data transmission, and intelligent diagnostic capabilities [57,58]. These systems enable real-time wound monitoring and, when integrated with patient-specific data and AI-based interpretative algorithms, can autonomously adjust dressing suitability and replacement timing. Additionally, when connected with telemedicine platforms, they offer a more effective, personalized therapeutic approach.

While the biomedical potential of nanoparticles is significant, their toxicity remains a critical consideration. The cytotoxicity of CeO_2_NPs is influenced by several factors, including particle size, synthesis method, cell type, concentration, exposure duration, and route of administration. However, at appropriate therapeutic doses, CeO_2_NPs are generally considered non-toxic in vivo [59]. For example, Zhao et al. demonstrated that alginate-based hydrogel wound dressings incorporating CeO_2_NPs significantly accelerated wound healing in a full-thickness skin wound model in rats, confirming their safety and efficacy [60]. Similarly, Attia et al. reported that standard therapeutic doses of CeO_2_NPs accumulated in the liver without inducing significant cytotoxic effects [61], while Cheng et al. found that 100 μg/mL CeO_2_NPs incorporated into a hydrogel did not significantly affect HaCaT cell viability as measured by the CCK-8 assay [43]. In the current study, PCL/PLA nanofibers containing 0.5% or 1% CeO_2_NPs were co-cultured with L929 and RAW264.7 cells for three days, and cell viability remained above 82%, indicating low cytotoxicity and excellent biocompatibility.

Despite the promising biological activities and encouraging experimental results, the clinical translation of CeO_2_NPs remains limited. To date, no clinical trials specifically targeting CeO_2_NPs have been conducted, underscoring the need for further preclinical and clinical investigations to validate their therapeutic efficacy and safety in human applications, particularly for wound healing and regenerative medicine.

## 4. Materials and Methods

### 4.1. Optimization of H. citrina Extraction Via Microwave-Assisted Extraction

*H. citrina* was obtained from a local food market in Taiwan. Extraction was performed using a microwave-assisted system (Ethos X, Milestone, Milano, Italy). A total of 250 g of *H. citrina* was added to 1000 mL of deionized water (1:4 *w*/*v*) to optimize the yield of total polyphenols by adjusting microwave power and extraction time.

The total polyphenol content was quantified as milligrams of gallic acid equivalent (GAE) per gram of extract, based on a standard calibration curve (y = 0.0036x + 0.0045, R^2^ = 0.9973). Total flavonoid content was expressed as milligrams of quercetin equivalent (QE) per gram of extract, determined using a quercetin calibration curve (y = 0.0004x + 0.0213, R^2^ = 0.9928).

### 4.2. Green Synthesis and Characterization of CeO_2_ Nanoparticles

To synthesize CeO_2_NPs, 3.72 g of Ce(NO_3_)_3_·6H_2_O was dissolved in 100 mL of *H. citrina* extract and stirred at 80 °C for 3 h. The resulting product was washed with 70% ethanol and calcined at 500 °C for 2 h.

Characterization was conducted using UV-Vis spectroscopy (BioMate 3, Thermo Fisher Scientific, Middleton, WI, USA), SEM/EDS (JSM-7610, JEOL, Tokyo, Japan), TEM (JJEM-2100F, JEOL, Tokyo, Japan), XRD (X’Pert Powder, PANalytical, Malvern, UK), and FTIR (JT/IR-4600, Jasco, Tokyo, Japan). Analyses included assessment of particle morphology, elemental composition, crystallinity, and functional groups.

The antioxidant capacity of *H. citrina* extract and CeO_2_NPs was evaluated Via the DPPH assay. Briefly, 2 mL of the sample was mixed with 2 mL of 2 × 10^−4^ M DPPH solution and incubated in the dark for 30 min. Absorbance at 517 nm was measured, with double-distilled water and vitamin C serving as negative and positive controls, respectively. The radical scavenging activity was calculated as follows:Scavenging effect (%) = [1 − (A_sample − A_blank)/A_control] × 100.(1)

### 4.3. Fabrication and Characterization of Aligned Electrospun Nanofibers

A polymer solution comprising 10 wt% PCL and 10 wt% PLA in hexafluoroisopropanol, with either 0.5 or 1.0 wt% CeO_2_NPs, was prepared and loaded into a 10 mL syringe fitted with an 18 G needle. Electrospinning was performed using a syringe pump onto a rotating collector (1200 rpm) covered with aluminum foil for 8 h under 25 kV, a 15 cm tip-to-collector distance, and a flow rate of 0.008 mL/min (JG-ESC 10, Falco Tech, Taipei, Taiwan). Fibers were subsequently dried under a fume hood for 24 h and stored in a desiccator.

Fiber morphology and diameter were assessed Via SEM. Mechanical properties (tensile strength, elongation, and Young’s modulus) were determined using a tensile testing machine (FL-508M1, FALCO, Taipei, Taiwan). Degradation rate, cerium ion release, and swelling ratio were evaluated. Ce ion release was quantified using ICP-OES (Optima 8300, PerkinElmer, Waltham, MA, USA). Swelling ratio was calculated as follows:Swelling ratio (%) = [(W_s − W_d)/W_d] × 100.(2)

Water vapor transmission rate (WVTR) was assessed based on a previous study [26], using a bottle setup with a 1.5 cm opening covered by the fiber mat and placed in an incubator at 37 °C for 24 h. WVTR of the fiber was calculated as follows:WVTR = Δm/(A × time).(3)
where Δm is the water loss weight (g), A is the area of the bottle mouth (m^2^), and time is the time (day).

### 4.4. Cytotoxicity and Cell Migration Assays of CeO_2_NPs

L929 and RAW264.7 cells (5 × 10^4^ cells/mL) were seeded in 96-well plates and incubated at 37 °C with 5% CO_2_ for 24 h. Cells were treated with 0, 0.5, and 1% (*w*/*v*) CeO_2_NPs in DMEM, filtered through a 0.22 μm membrane to ensure sterility, for 24, 48, and 72 h, followed by MTT assays. Absorbance was measured at 570 nm using a microplate reader (Multiskan™ FC, Thermo Scientific™, Middleton, WI, USA).

For the scratch wound assay, L929 cells (5 × 10^5^ cells/mL) were cultured in six-well plates for 24 h. A scratch was made using a 200 μL pipette tip, followed by treatment with 0, 0.5, and 1% (*w*/*v*) CeO_2_NPs. Cell migration was monitored at 12, 14, and 16 h post-treatment.

### 4.5. Anti-Inflammatory Activity and ROS Quantification

RAW264.7 cells (1 × 10^6^ cells/well) were seeded in six-well plates and treated with 0, 0.5, and 1% (*w*/*v*) CeO_2_NPs for 3 h before stimulation with IL-1β (100 ng/mL) for 24 h.

NO production was measured using the Griess assay. Cells were lysed with a micro-homogenizer and centrifuged. Supernatant (100 μL) was reacted with Griess reagents I and II and nitrite buffer (ab234044, Abcam, Cambridge, UK). Absorbance was measured at 570 nm and NO concentrations were calculated from a standard curve (y = 0.0065x + 0.0227, R^2^ = 0.9953).

For TNF-α analysis, ELISA kits were used according to the manufacturer’s instructions (ab208348, Abcam, Cambridge, UK), absorbance at 450 nm was detected by ELISA reader (Multiskan^TM^ FC, Thermo Scientific^TM^, Middleton, WI, USA), and the protein concentration was calculated by substituting into the standard curve [y = 0.57378ln(x) − 2.8128, R^2^ = 0.9917]. ROS levels were determined using a DCFDA Cellular ROS Detection Kit (ab113851, Abcam, Cambridge, UK) and measured at 485/535 nm.

### 4.6. Gene Expression Analysis via RT-PCR

Total RNA from RAW264.7 cells was isolated using TRIzol reagent (Thermo Fisher Scientific, Middleton, WI, USA). RNA concentration was quantified using a NanoDrop 2000 spectrophotometer. cDNA synthesis was performed using the SOLIScript^®^ RT synthesis kit (Solis BioDyne, Tartu, Estonia) with a Veriti^®^ Thermal Cycler. RT-PCR was conducted using HOT FIREPol^®^ EvaGreen^®^ qPCR Supermix with QuantStudio^TM^ 3 (Solis BioDyne, Tartu, Estonia). Expression levels were calculated using the 2^–ΔΔCt^ method and normalized to GAPDH. Primer sequences are listed in Table 1.

### 4.7. In Vivo Burn Wound Healing Assay

Male SD rats (7 weeks old, 226–250 g) were obtained from BioLasco Co., Ltd. (Taipei, Taiwan). All protocols were approved by the Institutional Animal Care and Use Committee of the National Defense Medical Center (approval no. IUCAC-23-043).

Burn wounds were induced by pressing a heated copper rod (9 mm diameter, 100 °C) onto the shaved dorsal skin for 10 s [5]. Wounds were treated with medical gauze (control), PCL/PLA nanofibers, or PCL/PLA/CeO_2_ nanofibers, then covered with 3D Tegaderm film and secured with elastic bandages. The wound area was photographed, and wound closure (%) was calculated as follows:Wound closure (%) = [(Initial area − Area on day n)/Initial area] × 100.(4)

Rats were euthanized on days 7 and 14. Tissue samples were fixed in formalin, embedded, sectioned, and stained with H&E and Masson’s trichrome to assess re-epithelialization, neovascularization, and collagen deposition.

### 4.8. Statistical Analysis

All experiments were performed in triplicate. Data are expressed as mean ± standard deviation (SD). Statistical analysis was conducted using IBM SPSS Statistics Base 23 software. One-way ANOVA followed by LSD post hoc tests were used to determine significance. A *p*-value of <0.05 or <0.01 was considered statistically significant and denoted as * or **, respectively.

## 5. Conclusions

*H. citrina* extract, which is rich in polyphenols and flavonoids, serves as an effective bioreducing and stabilizing agent for the green synthesis of CeO_2_NPs. The resulting CeO_2_NPs exhibited excellent biocompatibility with no detectable cytotoxicity, while also demonstrating enhanced cellular migration, antioxidant, and anti-inflammatory activities. Notably, they were also capable of modulating macrophage polarization, collectively supporting their suitability as bioactive agents for promoting wound healing. When incorporated into electrospun aligned nanofiber scaffolds, CeO_2_NPs significantly accelerated wound closure and tissue regeneration. Histopathological analysis further confirmed that CeO_2_NPs effectively reduced inflammatory cell infiltration, promoted re-epithelialization and granulation tissue formation, and enhanced collagen deposition. These findings provide compelling evidence for the therapeutic potential of CeO_2_NPs in wound healing applications and underscore their promise as a next-generation material for advanced wound dressings.

## Figures and Tables

**Figure 1 ijms-26-06087-f001:**
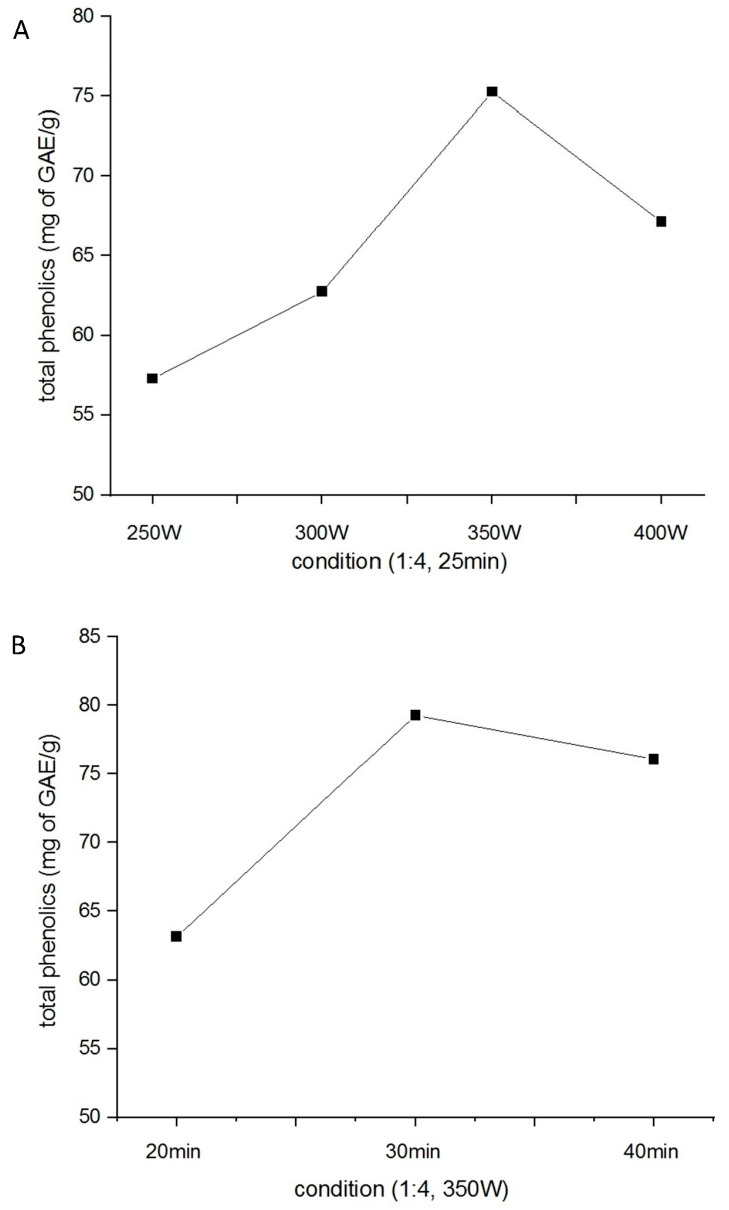
Optimization of the yield of *H. citrina* extract for highest total polyphenol, conducted at varying power levels (**A**) and different extraction durations (**B**).

**Figure 2 ijms-26-06087-f002:**
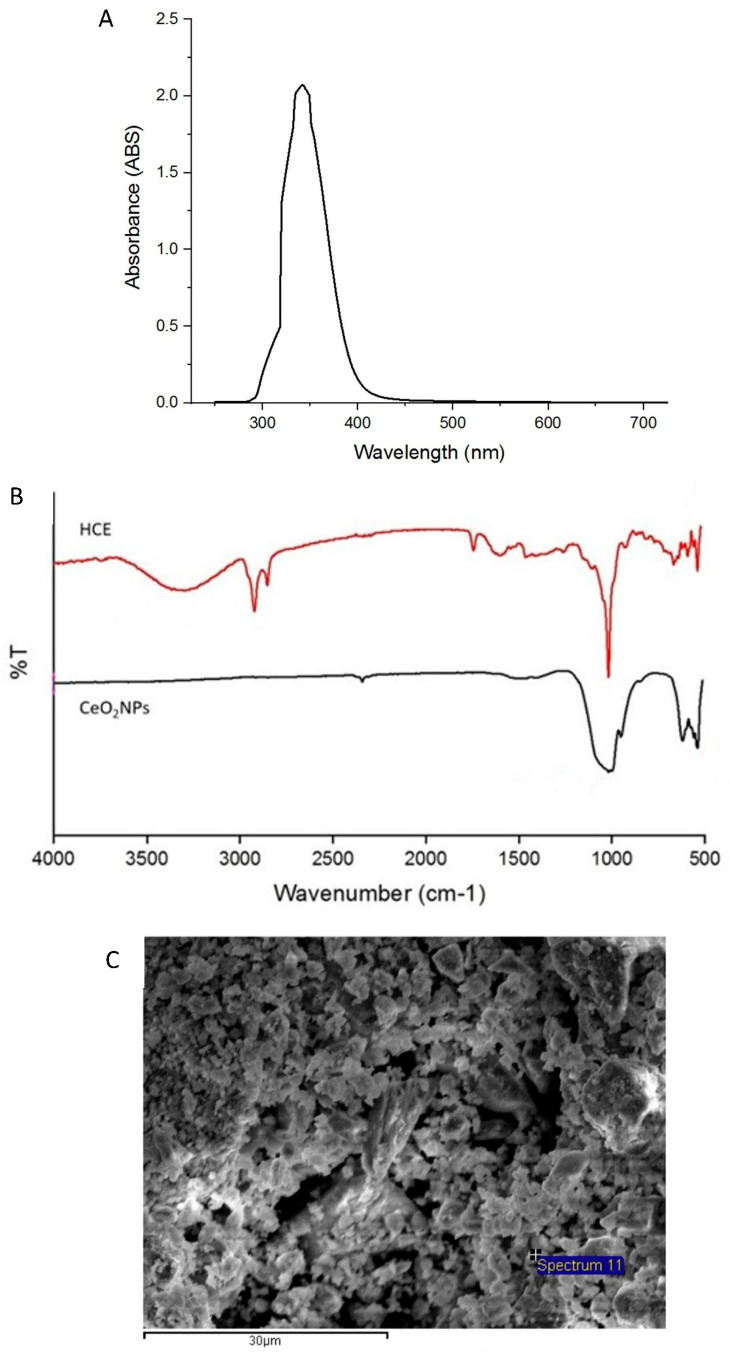
UV–Vis absorption spectrum (**A**), FTIR spectra (**B**), SEM image (**C**), EDS analysis (**D**), TEM image (**E**), particle size distribution (**F**), XRD pattern (**G**), SAED pattern (**H**), and antioxidant activity (**I**) of CeO_2_NPs. Values are expressed as mean ± SD (n = 3); * for comparison with PVA group; ** *p* < 0.01.

**Figure 3 ijms-26-06087-f003:**
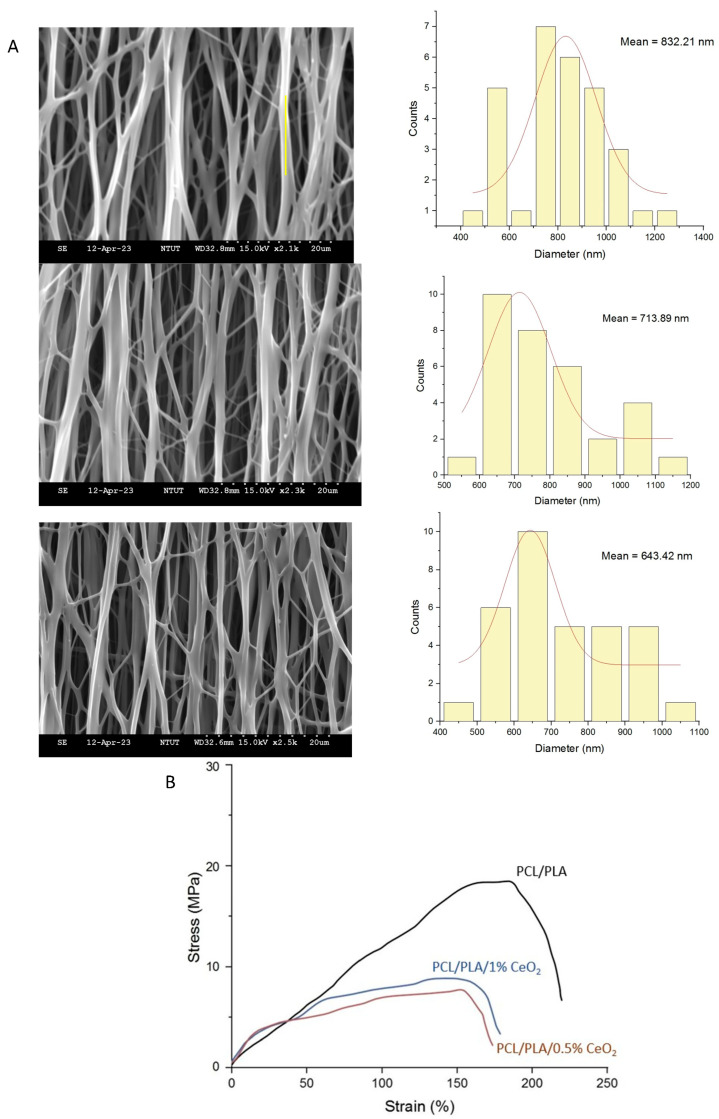
SEM images and diameter distribution (**A**), mechanical property (**B**), WVTR assay (**C**), degradation rate (**D**), swelling capacity (**E**), cerium ion release rate (**F**) of PCL/PLA/CeO_2_NPs nanofibers. Values are expressed as mean ± SD (n = 3), * for comparison with PVA group; * *p* < 0.05, ** *p* < 0.01.

**Figure 4 ijms-26-06087-f004:**
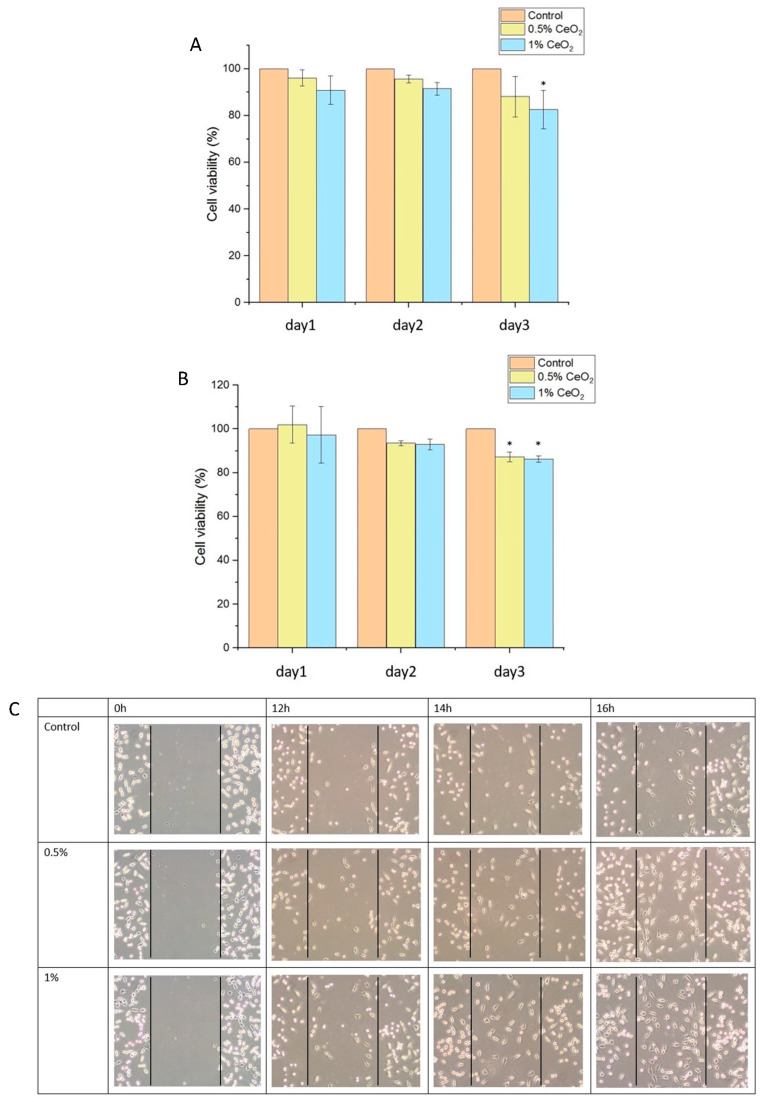
Cytotoxicity of CeO_2_NPs to L929 (**A**) and RAW264.7 cells (**B**), cell migration between the scratch of 200 μL pipette tip, 100× (**C**) by CeO_2_NPs-treated L929 cells and quantification (**D**). Values are expressed as mean ± SD (n = 3), * for comparison with PVA group; * *p* < 0.05, ** *p* < 0.01.

**Figure 5 ijms-26-06087-f005:**
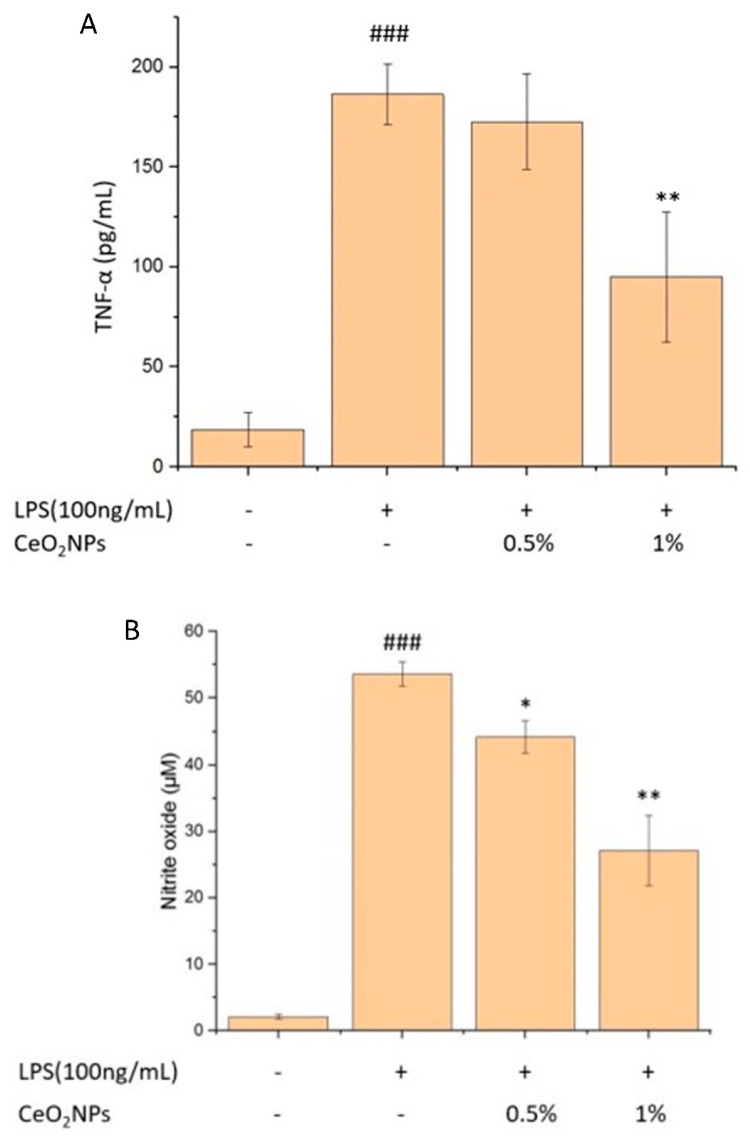
TNF-α expression (**A**), NO expression (**B**), and ROS content (**C**) of LPS-induced, CeO_2_NPs-pretreated RAW264.7 cells. # for comparison with control group, ### *p* < 0.001 and * for comparison with LPS-treated group; * *p* < 0.05, ** *p* < 0.01.

**Figure 6 ijms-26-06087-f006:**
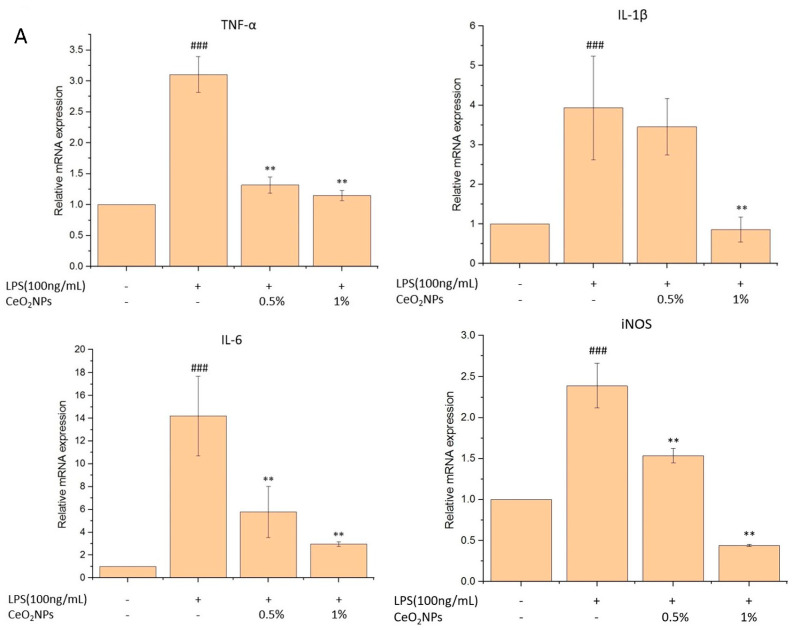
The gene expression of TNF-α, IL-6, IL-1β, and iNOS (**A**), as well as IL-10, IL-4, Arg-1, and TGF-β (**B**) of LPS-induced, CeO_2_NPs-pretreated RAW264.7 cells. # for comparison with control group, ### *p* < 0.001, and * for comparison with LPS-treated group; * *p* < 0.05, ** *p* < 0.01.

**Figure 7 ijms-26-06087-f007:**
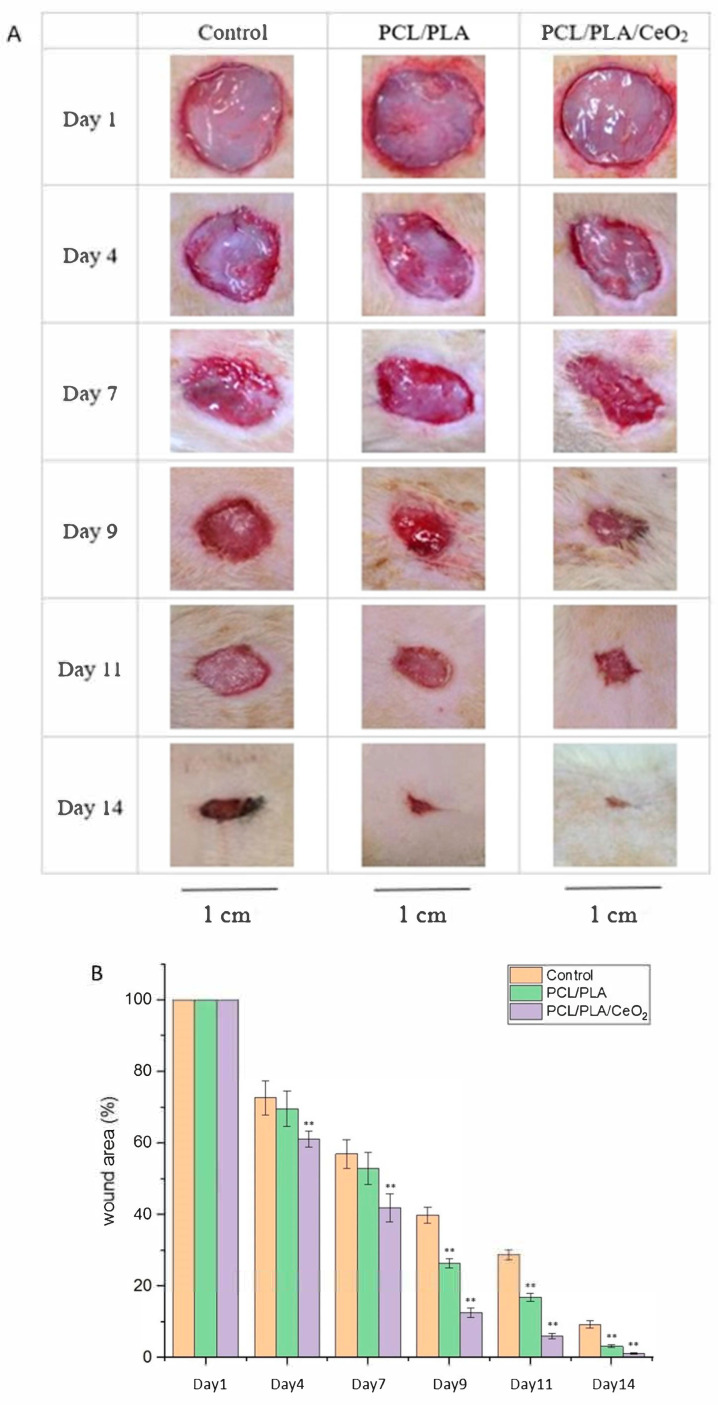
Photographs of wounds (**A**) and closure of the wound area (**B**); H&E staining, 40× (**C**); quantification of granulation tissue area and the number of newly formed blood vessels (**D**); Masson’s trichrome staining of the wound tissue, 12.5× (**E**); quantification of collagen deposition (**F**) under treatment with or without PCL/PLA/1%CeO_2_ fibers. Values are expressed as mean ± SD (n = 3). Values are expressed as mean ± SD (n = 3), * for comparison with day 1 group; ** *p* < 0.01.

**Table 1 ijms-26-06087-t001:** The used PCR primers for qRT-PCR.

Oligo Name	Oligo Seq (5′-3′)
IL-6 forward	TACCACTTCACAAGTCGGAGGC
IL-6 reverse	CTGCAAGTGCATCATCGTTGTTC
IL-1β forward	GAAATGCCACCTTTTGACAGTG
IL-1β reverse	TGGATGCTCTCATCAGGACAG
TNF-α forward	CCTGTAGCCCACGTCGTAG
TNF-α reverse	GGGAGTAGACAAGGTACAACCC
iNOS forward	GTTCTCAGCCCAACAATACAAGA
iNOS reverse	GTGGACGGGTCGATGTCAC
Arg-1 forward	CTCCAAGCCAAAGTCCTTAGAG
Arg-1 reverse	AGGAGCTGTCATTAGGGACATC
IL-4 forward	GGTCTCAACCCCCAGCTAGT
IL-4 reverse	GGTCTCAACCCCCAGCTAGT
IL-10 forward	TACAGCCGGGAAGACAATAA
IL-10 reverse	AAGGAGTCGGTTAGCAGTAT
TGF-β forward	GTCCTTGCCCTCTACAACCA
TGF-β reverse	GTTGGACAACTGCTCCACCT
GAPDH forward	AGGTCGGTGTGAACGGATTTG
GAPDH reverse	GGGGTCGTTGATGGCAACA

## Data Availability

The original contributions presented in this study are included in the article. Further inquiries can be directed to the corresponding author.

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
