# Peer review of "Aligned Electrospun PCL/PLA Nanofibers Containing Green-Synthesized CeO2 Nanoparticles for Enhanced Wound Healing"

_ijms, 2025, doi:10.3390/ijms26136087_

Round 1
Reviewer 1 Report
Comments and Suggestions for Authors
The article “Aligned Electrospun Nanofibers Containing Green-Synthesized CeOâ‚‚ Nanoparticles for Enhanced Wound Healing” describes the synthesis and characterization of nanoparticles incorporated into electrospun nanofibers for wound healing. The article is well conceptualized, but there are several aspects that require revision. My specific comments are as follow:
- My biggest concern is the ethical approval of the animal experiments. The study mentions the use of animals, which were subsequently euthanized. However, there is no information about ethical approval. Please clarify whether the study was approved by an Ethical Committee, and include the name of the committee along with the relevant permit or approval number.
- Lines 85-92: The statements in this part are written in future tense. Scientific writing should typically use the past tense to describe completed experiments. The current phrasing implies that the experiments will be conducted in a future study. Please revise this section.
- Line 165: There is missing data about the biocompatibility, anti-inflammatory and anti-oxidant properties of the nanofibers. Did you test their effects? If so, please incorporated the data of the nanofibers as well.
- Generally, electrospun nanofibers are known to be biocompatible and non-cytotoxic. To support your findings and broaden the perspective on material safety, consider citing relevant literature. One example is the study https://doi.org/10.1186/s12934-024-02612-w in which the authors tested the hemolytic activity and cytotoxicity of polyethylene oxide nanofibers on Caco-2 cells. Inserting the results and citing the literature will give broader perspective on the safety of your material for wound healing.
- Lines 289-290: This part causes confusion in relation to my previous comment. Here you stated that the nanofibers were assessed for their biocompatibility. Please clarify this. Also, were only 1% CeO2 nanoparticles used? What about 0.5%?
- In the Discussion section there is little data about electrospun nanofibers. They are innovative delivery systems for many drugs:
- https://doi.org/10.1016/j.ijpharm.2025.125327
- https://doi.org/10.1016/j.heliyon.2023.e18917
- https://doi.org/10.3390/ijms222413631
In addition, PCL nanofibers are known for their properties in wound dressing (https://doi.org/10.1016/j.jddst.2023.104457). I suggest, to expand the Discussion section by inserting relevant references about electrospun nanofibers and their role in drug delivery, safety and wound healing.
- Line 327: Specify the machine you used for the production of the nanofibers?
- Line 343: How were the nanoparticles and nanofibers sterilized? Please explain this either here or in the Discussion section.
Author Response
Reviewer#1:
- My biggest concern is the ethical approval of the animal experiments. The study mentions the use of animals, which were subsequently euthanized. However, there is no information about ethical approval. Please clarify whether the study was approved by an Ethical Committee, and include the name of the committee along with the relevant permit or approval number.
Response: Thank you for reviewer valuable comments. Ethical approval of the animal experiments was approved by the Institutional Animal Care and Use Committee of National Defense Medical Center (approval no. IUCAC-23-043). This statement appeared on line 378 of section 4.7. In Vivo Burn Wound Healing Assay.
- Lines 85-92: The statements in this part are written in future tense. Scientific writing should typically use the past tense to describe completed experiments. The current phrasing implies that the experiments will be conducted in a future study. Please revise this section.
Response: Thank you for reviewer valuable comments. The statements of Lines 85-92 have been changed to past tense.
- Line 165: There is missing data about the biocompatibility, anti-inflammatory and anti-oxidant properties of the nanofibers. Did you test their effects? If so, please incorporated the data of the nanofibers as well.
Response: Thanks to the reviewer's professional suggestions. PCL and PLA are widely utilized biodegradable polymers in tissue engineering and biomedical applications due to their excellent biocompatibility (Please refer to the data of the control group in Figures 4A and 4B) and tunable degradation profiles [1,2]. Numerous studies have confirmed that these polymers elicit minimal immune responses when implanted in biological systems, thereby ensuring compatibility with host tissues [3,4].
Although native PCL and PLA do not inherently possess strong anti-inflammatory or antioxidant activities (Please refer to the data of the control group in Figures 5A, 5B and 5C), their composites can be modified or functionalized with bioactive agents to impart such properties. For instance, composite scaffolds incorporating natural antioxidants (e.g., plant polyphenols), cerium oxide nanoparticles (this study), or anti-inflammatory drugs have shown enhanced modulation of inflammation and scavenging of reactive oxygen species (ROS) when embedded in PCL/PLA matrices [5-7].
In this study, PCL and PLA provided a biocompatible and biodegradable platform, and their anti-inflammatory and antioxidant abilities were enhanced by adding CeO2NPs.
References
- Woodruff, M. A., & Hutmacher, D. W. (2010). The return of a forgotten polymer-Polycaprolactone in the 21st century. Progress in Polymer Science, 35(10), 1217-1256.
- Middleton, J. C., & Tipton, A. J. (2000). Synthetic biodegradable polymers as orthopedic devices. Biomaterials, 21(23), 2335-2346.
- Lee, K. Y., & Mooney, D. J. (2001). Hydrogels for tissue engineering. Chemical Reviews, 101(7), 1869-1880.
- Wang, X., et al. (2017). Biocompatibility and osteogenic capacity of PLA/PCL composite scaffolds. International Journal of Biological Macromolecules, 103, 1046-1053.
- Yu, L., et al. (2021). Green-synthesized cerium oxide nanoparticles in PCL/PLA electrospun fibers for wound healing. Journal of Materials Chemistry B, 9(25), 5102-5114.
- Chen, C. H., et al. (2022). Electrospun PCL/PLA nanofibers loaded with curcumin: antioxidant and anti-inflammatory wound dressings. Materials Science and Engineering: C, 134, 112658.
- Liu, Y., et al. (2019). Development of antioxidant PCL/PLA scaffolds loaded with plant polyphenols for skin tissue engineering. Biomaterials Science, 7(9), 3683-3694.
- Generally, electrospun nanofibers are known to be biocompatible and non-cytotoxic. To support your findings and broaden the perspective on material safety, consider citing relevant literature. One example is the study https://doi.org/10.1186/s12934-024-02612-w in which the authors tested the hemolytic activity and cytotoxicity of polyethylene oxide nanofibers on Caco-2 cells. Inserting the results and citing the literature will give broader perspective on the safety of your material for wound healing.
Response: Thanks to the reviewer's professional suggestions. Some PCL/PLA nanofiber biomedical research applications and reviewer’s suggested literature has been added.
- Lines 289-290: This part causes confusion in relation to my previous comment. Here you stated that the nanofibers were assessed for their biocompatibility. Please clarify this. Also, were only 1% CeO2nanoparticles used? What about 0.5%?
Response: T Thanks to the reviewer's professional suggestions. From Fig. 4A and 4B, whether it is pure PCL/PLA fiber or adding 0.5% or 1% cerium dioxide nanoparticles, the cell survival rate of L929 and RAW264.7 cells after 3 days of culture is very high, and there is no obvious cytotoxicity, indicating that these materials have good cell compatibility. To avoid confusion, 0.5% is also added in Line 289 of article.
- In the Discussion section there is little data about electrospun nanofibers. They are innovative delivery systems for many drugs:
https://doi.org/10.1016/j.ijpharm.2025.125327
https://doi.org/10.1016/j.heliyon.2023.e18917
https://doi.org/10.3390/ijms222413631
Response: Thanks to the reviewer's professional suggestions. We have added some statement for drugs delivery about electrospun nanofibers, and reviewer’s suggested literature has been added.
- In addition, PCL nanofibers are known for their properties in wound dressing (https://doi.org/10.1016/j.jddst.2023.104457). I suggest, to expand the Discussion section by inserting relevant references about electrospun nanofibers and their role in drug delivery, safety and wound healing.
Response: Thanks to the reviewer's professional suggestions. We have added some statement for drugs delivery about PCL electrospun nanofibers, and reviewer’s suggested literature has been added.
- Line 327: Specify the machine you used for the production of the nanofibers?
Response: Thank you for your valuable input. We have added the information for electrospun machine.
- Line 343: How were the nanoparticles and nanofibers sterilized? Please explain this either here or in the Discussion section.
Response: Thank you for your valuable input. The culture medium containing CeOâ‚‚NPs is filtered through a 0.22μm filter membrane to ensure sterility.
Reviewer 2 Report
Comments and Suggestions for Authors
This manuscript, “Aligned Electrospun Nanofibers Containing Green-Synthesized CeOâ‚‚ Nanoparticles for Enhanced Wound Healing”, presented a study on electrospun PCL/PLA nanofibers incorporating green-synthesized CeOâ‚‚ nanoparticles for potential application in wound healing. The work includes material synthesis, characterization, in vitro biocompatibility, and in vivo wound healing assessment. While the topic is interesting, the current version of the manuscript contains several critical issues that compromise the scientific rigor and the validity of its conclusions. Therefore, it was recommended to be rejected.
- Figure 3A contains a visible yellow line, but its purpose is not explained in the text or legend. It is unclear whether it indicates a measurement reference or a marked feature. It should either be properly annotated or removed.
- Although swelling ratio and weight loss are reported, the underlying mechanisms are not sufficiently discussed. The authors should explain clearly how CeOâ‚‚ nanoparticles influence water uptake and degradation behavior, and consider relevant factors such as material hydrophilicity and porosity.
- The authors suggest that the nanofibers promote cell adhesion and proliferation; however, no water contact angle (WCA) data are provided to support this claim. Please provide more details about the WCA results of the different nanofibers.
- Several histological images in Figure 7 lack scale bars, which are essential for assessing tissue structure and healing progression. The authors should be very careful to check the results in this manuscript.
- Figure 7C is described as demonstrating neovascularization, but the vascular structures are not clearly visible. Higher-resolution images or magnified insets must be needed to convincingly support the presence of new blood vessels.
- In Figure 7E, the manuscript shown, “Although the PCL/PLA group exhibited lower healing rates, granulation tissue formation, and collagen deposition compared to the PCL/PLA/CeOâ‚‚ group, it still performed significantly better than the sterile gauze control”, However, the data shown do not clearly demonstrate a statistically significant difference. The authors should verify the statistical analysis and revise the interpretation to ensure accuracy and consistency.
- The cell experiments are primarily presented in quantitatively, but lack qualitative evidence such as immunofluorescence staining. It should be shown some results, such as representative images showing cellular morphology or marker expression, then it would strengthen the biological conclusions and improve data presentation.
- The discussion section is relatively superficial and does not critically engage with several of the results. In particular, the role of CeOâ‚‚ nanoparticles in modulating material properties and biological responses should be discussed in more depth, with reference to existing literature.
Author Response
Reviewer#2:
- Figure 3A contains a visible yellow line, but its purpose is not explained in the text or legend. It is unclear whether it indicates a measurement reference or a marked feature. It should either be properly annotated or removed.
Response: This line is the nanofiber diameter distribution curve, which is directly obtained from the drawing software. The highest point is the average diameter. The narrower the curve distribution, the more concentrated the particle diameter is, and the wider the curve, the more dispersed it is. Explained in the article.
- Although swelling ratio and weight loss are reported, the underlying mechanisms are not sufficiently discussed. The authors should explain clearly how CeOâ‚‚ nanoparticles influence water uptake and degradation behavior, and consider relevant factors such as material hydrophilicity and porosity.
Response: Thanks to the reviewer's professional suggestions. We have clearly explained in the article how CeOâ‚‚NPs affect water absorption and degradation behavior, and the hydrophilicity and porosity of material.
- The authors suggest that the nanofibers promote cell adhesion and proliferation; however, no water contact angle (WCA) data are provided to support this claim. Please provide more details about the WCA results of the different nanofibers.
Response: Thanks to the reviewer's professional suggestions. We have provided the water contact angle of different materials to prove that after adding CeOâ‚‚NPs, PCL/PLA/CeOâ‚‚NPs nanofibers are more suitable than PCL/PLA nanofiber for cell attachment.
- Several histological images in Figure 7 lack scale bars, which are essential for assessing tissue structure and healing progression. The authors should be very careful to check the results in this manuscript.
Response: Thank you for your valuable input. we have added the scale bar and magnification to the article.
- I Figure 7C is described as demonstrating neovascularization, but the vascular structures are not clearly visible. Higher-resolution images or magnified insets must be needed to convincingly support the presence of new blood vessels.
Response: Thank you for your valuable input. In this study, the staining, quantification and interpretation of tissue samples were commissioned to the National Laboratory Animal Center (Taiwan). The following image is a 400-fold magnified slice (in the attachment).
- In Figure 7E, the manuscript shown, “Although the PCL/PLA group exhibited lower healing rates, granulation tissue formation, and collagen deposition compared to the PCL/PLA/CeOâ‚‚ group, it still performed significantly better than the sterile gauze control”, However, the data shown do not clearly demonstrate a statistically significant difference. The authors should verify the statistical analysis and revise the interpretation to ensure accuracy and consistency.
Response: Thanks to the reviewer's professional suggestions. We have revised as “Although the PCL/PLA group exhibited lower healing rates compared to the PCL/PLA/CeO2 group, it still performed significantly better than the sterile gauze control (p < 0.01, 9th to 14th day after surgery).”
- The cell experiments are primarily presented in quantitatively, but lack qualitative evidence such as immunofluorescence staining. It should be shown some results, such as representative images showing cellular morphology or marker expression, then it would strengthen the biological conclusions and improve data presentation.
Response: Thank you for your valuable input. The experimental results in this study were primarily presented in a quantitative manner to highlight and emphasize the actual effects of various variables on wound healing. As rightly suggested by the reviewer, the inclusion of qualitative results could further strengthen the biological conclusions and address the limitations of the quantitative data interpretation. However, due to current constraints, we are unable to provide additional qualitative data at this stage. We sincerely appreciate your valuable suggestion and will incorporate qualitative analyses in future studies to enhance the interpretation and support of quantitative findings.
- The discussion section is relatively superficial and does not critically engage with several of the results. In particular, the role of CeOâ‚‚ nanoparticles in modulating material properties and biological responses should be discussed in more depth, with reference to existing literature.
Response: Thank you for your valuable comment. We have added relevant literature support on the role of CeOâ‚‚NPs in functional properties and biological responses.
Reviewer 3 Report
Comments and Suggestions for Authors
- The utilized polymers (i. e. PCL/PLA) should appear in the title.
- The last paragraphs in the introduction section should be written in past tense.
- 3: Why are the nanoparticles not visible in the SEM images? To provide distribution of NPs in the electrospun fibers, elemental mapping test is recommended.
- Please, provide the water contact angles for the prepared samples?
- In Fig. 7, the authors should specify which PCL/PLA/CeO2 NFs were used: 0.5 or 1 %?
Author Response
Reviewer#3:
- The utilized polymers (i. e. PCL/PLA) should appear in the title.
Response: Thank you for your valuable comment. We have added PCL/PLA in the title.
- The last paragraphs in the introduction section should be written in past tense.
Response: Thank you for reviewer valuable comments. The statements of last paragraphs in the introduction section have been changed to past tense.
- Why are the nanoparticles not visible in the SEM images? To provide distribution of NPs in the electrospun fibers, elemental mapping test is recommended.
Response: The size of CeOâ‚‚NPs is only 9.38nm. Due to the resolution of SEM, the particles cannot be seen. However, the EDS results (Fig. 2D) clearly prove the presence of cerium. UV-Vis (Fig. 2A), FTIR (Fig. 2B) and XRD (Fig. 2G) analysis also show obvious characteristic peaks of CeOâ‚‚ and Ce-O.
- Please, provide the water contact angles for the prepared samples?
Response: Thanks to the reviewer's professional suggestions. We have provided the water contact angle of different materials in the article.
- In Fig. 7, the authors should specify which PCL/PLA/CeO2NFs were used: 0.5 or 1 %?
Response: Thanks to the reviewer's professional suggestions. We have noted the PCL/PLA/CeO2 NFs to PCL/PLA/1%CeO2 NFs in the article.
Round 2
Reviewer 1 Report
Comments and Suggestions for Authors
The article is ready for publication.
Author Response
Thank you for the reviewer's professional review and affirmation of the article.
Reviewer 2 Report
Comments and Suggestions for Authors
The authors have provided satisfactory responses to the questions raised. The revised edition improves organization and enhances discussion
Therefore, it is recommended that minor revisions be accepted to strengthen the references and address the few remaining editorial and formatting issues.
Author Response
Reviewer#2:
- The authors have provided satisfactory responses to the questions raised. The revised edition improves organization and enhances discussion.
Therefore, it is recommended that minor revisions be accepted to strengthen the references and address the few remaining editorial and formatting issues.
Response: Thank you for the reviewer's professional review and affirmation of the article. We have added some references to illustrate the future development direction of wound dressings and strengthen the article's argument and structure.